

**Black carbon content of traffic emissions impacts significantly on black carbon**
**mass size distributions and mixing states**
**Fei Li[1,3,5#], Biao Luo [2,4#], Miaomiao Zhai[2,4], Li Liu[3], Gang Zhao[7], Hanbing Xu[6], Tao Deng[3],**
**Xuejiao Deng[3], Haobo Tan[3], Ye Kuang[2,4*], Jun Zhao[1*]**
[1] School of Atmospheric Sciences, Guangdong Province Key Laboratory for Climate Change and
Natural Disaster Studies, and Southern Marine Science and Engineering Guangdong Laboratory
(Zhuhai), Sun Yat-sen University, Zhuhai, 519082, China
[2] Institute for Environmental and Climate Research, Jinan University, Guangzhou, 511443, China.
[3] Institute of Tropical and Marine Meteorology, China Meteorological Administration, Guangzhou,
510640,China
[4] Guangdong-Hongkong-Macau Joint Laboratory of Collaborative Innovation for Environmental
Quality, Guangzhou, 511443, China.
[5] Xiamen Key Laboratory of Straits Meteorology, Xiamen Meteorological Bureau, Xiamen, 361012,
China
[6] Experimental Teaching Center, Sun Yat-Sen University, Guangzhou 510275, China
[7] State Key Joint Laboratory of Environmental Simulation and Pollution Control, International Joint
Laboratory for Regional Pollution Control, Ministry of Education, College of Environmental Sciences
and Engineering, Peking University, Beijing, 100871, China
# These authors contribute equally to this paper.
Corresponding author: Ye Kuang (kuangye@jnu.edu.cn) and Jun Zhao
(zhaojun23@mail.sysu.edu.cn)



**Abstract**

Both the size and mixing state of black carbon (BC)-containing aerosols are crucial in estimating the environmental, health and climate impacts of BC. Traffic emissions are a major global source of BC, however, parameterization of BC mass size distributions and mixing states associated with traffic remains lacking due to its dependence on vehicle types and driving conditions. To investigate BC mass size distributions and mixing states associated with traffic emissions, a field campaign was conducted in Guangzhou urban area during winter, which used a system coupling a differential mobility analyzer (DMA) and single-particle soot photometer (SP2) to measure BC mass size distributions in the range of 100 to 700 nm. The resolved primary organic aerosols were hydrocarbon-like organic aerosols (HOA) and cooking-like organic aerosols (COA), refractory BC (rBC) which was detected by the DMA-SP2 and correlated highly with HOA ($R^2$=0.88), confirming that traffic emissions are the dominant source of atmospheric BC during the observations. The BC mass size distribution was found to be best fitted by a lognormal distribution, with a geometric mean ($D_{g,BC}$) of 258±16 nm, varying between 200 and 300 nm. During daytime, active formation of secondary nitrate and organic aerosols was observed, but it had little effect on the variations of BC mass size distributions. Further analyses revealed that $D_{g,BC}$ was highly correlated with rBC/HOA (R=0.66) in a linear form of $D_{g,BC}$= 34×rBC/HOA+177, demonstrating that the BC content of traffic emissions significantly impacts the BC mass size distributions. In addition, the size-dependent fractions of BC-containing aerosols in all types of aerosols ($f_{BCc}$) and the fraction of identified externally mixed (bare/thinly coated) BC particles in all BC-containing aerosols ($f_{ext}$) were also characterized. It was found that the daytime secondary aerosol formation reduced both $f_{BCc}$ and $f_{ext}$, with the decrease of $f_{ext}$ being more pronounced for larger particles, possibly due to the higher relative coating thickness. Variations in $f_{ext}$ during nighttime were mainly controlled by the emission conditions. For example, $f_{ext}$ for 600 nm particles decreased from 0.82 to 0.46 as rBC/HOA increased from 1 to 3.5 while the mass ratios of secondary aerosols to rBC varied little, demonstrating that the BC content also significantly affects the mixing states of freshly emitted BC from traffic emissions. This study suggests that BC content likely plays a key role in parameterizing both mass size distributions and mixing states of BC from traffic emissions and hence has significant implications for accurate representation of BC from different sources when modeling the impacts of BC.



## 1 Introduction

Aerosols impact significantly on human health through deposition on human tissues, visibility, weather and climate through interacting with solar radiation and acting as cloud condensation nuclei (CCN). Most of the atmospheric aerosols scatter readily while absorbing negligibly or little solar irradiation, but one exception is black carbon (BC) which absorbs solar irradiation strongly and thus heats the atmosphere. This strong absorption makes BC the second atmospheric warming component (Bond et al., 2013) and plays a major role in climate and air pollution. Menon et al. (2002) found that BC's absorption could affect trends of droughts and floods in India and China by altering regional atmospheric stability and vertical motions. The heating effects of BC in the atmospheric boundary layer can suppress boundary layer turbulence, impacting boundary layer development and meteorology (Wilcox et al., 2016), and consequently affecting local haze formation (Ding et al., 2016). Moreover, BC-containing aerosols can interact with clouds and serve as CCN (Zhang et al., 2017;Motos et al., 2019;Hu et al., 2021), thus indirectly impacting climate (Koch and Del Genio, 2010). These effects of BC can be quantitatively determined through its radiative forcing. In addition, inhalation of atmospheric BC also poses a threat to human health; evidence suggests that it may be associated with changes in subclinical cardiovascular health effects in individuals (Nichols et al., 2013).

The radiative and health effects of BC are highly dependent on its size and mixing state, as these factors determine the distribution of BC in aerosols and their optical properties (Bond et al., 2006), hygroscopic growth (Liu et al., 2013), and depositions in the human respiratory tract (Man et al., 2022). Freshly emitted BC can be either almost bare or coated with other materials, usually organic aerosols, which can then undergo rapid aging processes through vapor condensation or serving as medium for reactions (Zhang et al., 2018a;Zhang et al., 2020a;Zhang et al., 2021). This leads to size and morphological changes of BC-containing aerosols (Zhang et al., 2008) and influences their physical properties, such as hygroscopicity (Liu et al., 2013) and activation abilities (Ding et al., 2019;Yu et al., 2022). The coating of other components on BC can also significantly affect the optical properties of BC-containing aerosols, such as the lensing effect that enhances light absorption (Bond et al., 2006;Peng et al., 2016). This effect is non-linear (Liu et al., 2017;Wang et al., 2021) and is highly dependent on the mixing state heterogeneity of BC-containing aerosols (Fierce et al., 2020;Zhao et al., 2021;Zhai et al., 2022a). In addition, Zhao et al. (2019) showed that the BC mass size distributions



also play a major role in the direct radiative effects of BC. The size and mixing states of BC, as well
as the chemical composition of its coatings, vary significantly with sources (Zhang et al., 2020b). This
results in marked differences in the aging processes of freshly emitted BC from different sources in
the atmosphere, depending on the emission sources and meteorological conditions in a given location.

Guangzhou is an expansive metropolis in the highly industrialized Pearl River Delta (PRD) region

of China. Previous studies have shown that emissions from fossil fuel combustion are major sources
of BC (Liu et al., 2014) while biomass burning emissions might also make certain contributions during
autumn and winter (Sun et al., 2020). However, recent studies on source apportionment have not
detected obvious signals attributed to biomass burning in autumn and winter, suggesting that other
sources such as traffic activities are the main contributors to BC emissions (Guo et al., 2020;Chen et
al., 2021b;Liu et al., 2022;Zhai et al., 2022b). Few studies have used the single-particle soot
photometer (SP2) to measure bulk BC mass concentrations and mixing states in Guangzhou urban area
(Huang et al., 2011; Tao et al., 2021). Furthermore, no comprehensive measurements that characterize
both size distribution and mixing states of BC in this region have been conducted, and the factors that
control variations in BC mass size distributions and mixing states remain unknown. BC emissions
from diesel vehicles dominate traffic BC emissions (Bond et al., 2013) and depend on many factors,
such as fuel type, engine operating conditions, engine types, driving patterns, and environmental
conditions (Adler et al., 2010). These conditions have a significant impact on the size distributions and
mixing states of emitted particles (Lähde et al., 2011;Xu et al., 2014), therefore BC mass size
distributions and mixing states vary a lot in real traffic conditions. While numerous studies have
examined the BC size and mixing states of emissions from various types of vehicles (Adler et al.,
2010;Liu et al., 2017), only a few have directly investigated the BC size distribution and mixing states
as a function of aerosol mobility diameter using the DMA-SP2 system which couples differential
mobility analyzer (DMA) and SP2 (Raatikainen et al. (2017), and reported the average coating
characteristics of aerosol particles emitted from diesel vehicle exhaust (Han et al., 2019;Zhang et al.,
2020b). However, the variations of BC mass size distributions and BC mixing states from real traffic
emissions using the DMA-SP2 system have rarely been studied, and how to parameterize them remains
elusive. This study carried out a field campaign and employed the DMA-SP2 system to investigate the
dominant contribution of traffic emissions to atmospheric BC, which provided an ideal opportunity to
evaluate how primary traffic emissions and their subsequent aging can affect BC mass size





distributions and mixing states.

## 2 Materials and Methods

### 2.1 Campaign information

The campaign was conducted to characterize BC mass size distributions and mixing states from
11 January to 27 February 2022 at the Haizhu wetland park in Guangzhou. The instruments used for
characterizing aerosol chemical and physical properties included a quadrupole aerosol chemical
speciation monitor (Q-ACSM) for monitoring aerosol chemical compositions, a DMA-SP2 system for
measuring BC mass size distributions and mixing states, and a scanning mobility particle sizer (SMPS)
system for measuring aerosol particle number size distributions ranging from 13 nm to 800 nm. An
AE33 aethalometer (Drinovec et al., 2015) was used to measure aerosol absorptions at multiple
wavelengths and indirectly measure bulk BC mass concentrations. A PM$_{2.5}$ inlet (BGI, SCC 2.354)
with a flow rate of 8 L/min was used for aerosol sampling. The flow rates of the Q-ACSM, CPC, SP2
and AE33 instruments were set to 3 L/min, 0.3 L/min, 0.1 L/min, and 5 L/min, respectively, to meet
the flow rate requirement of the impactor although there are some deviations. All instruments were
housed in a temperature-controlled container (23-27 ℃) and placed downstream of a Nafion drier
designed to lower the sample RH to less than 35% (placed outside of the container and vertically to
ensure a straight line of the sampling route so that sampling loss of aerosols can be minimized).
Meteorological parameters such as temperature, wind speed and direction, and relative humidity (RH)
were measured using an automatic weather station. Further details about this site can be found in Liu
et al. (2022).

### 2.2 DMA-SP2 system and data processing

The SP2 (Droplet Measurement Technologies) can measure aerosol scattering and incandescence
signals of individual particles and identify if they contain detectable BC mass. It can also provide BC
mass concentrations at the single particle level, thus allowing for the determination of BC mixing states.
The scattering signals can be used to estimate the particle size of each BC-free particle; however, a
leading-edge-only method is required for sizing  BC-containing particles (Schwarz et al., 2006), with
the estimated optical-equivalent size potentially deviating substantially from the mobility size due to
variations in aerosol refractive index and morphology. As such, the use of the DMA-SP2 system to
measure BC mass size distributions and mixing states has been previously proposed (Raatikainen et



al., 2017;Han et al., 2019;Sarangi et al., 2020;Zhao et al., 2021), which complement the size
measurements and additional size information can be used to derive physical properties such as
morphology and effective densities (Zhang et al., 2018b;Wu et al., 2019). A similar system has also
been used for other applications, such as investigation of the hygroscopic properties of BC-containing
aerosols (McMeeking et al., 2011; Liu et al., 2013). The DMA-SP2 set-up of Zhao et al. (2021)
employed a continuous scanning mode of the DMA, allowing for black carbon mass size distribution
(BCMSD) measurements with a time resolution of 5 minutes. However, accurate matching of the time
of particles in the DMA and SP2 is necessary. Some previous studies passed size-selected
monodisperse aerosols to the SP2 at only a few diameters (Zhang et al., 2018b;Han et al., 2019),

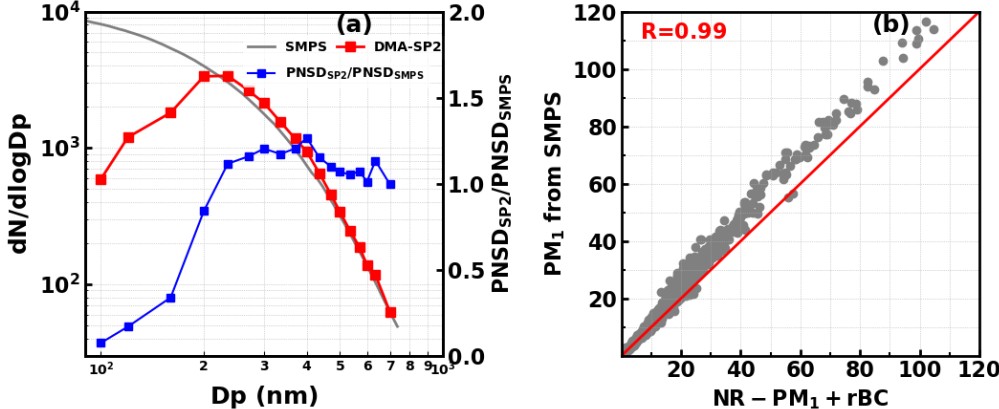

Figure 1. (a) Comparisons between average PNSD observed by the SMPS and inverted from the DMA-SP2 system; (b) Comparison between NR-PM$_1$+rBC and PM$_1$ mass concentrations calculated from SMPS measurements.

limiting the retrieval of BC mass size distributions and mixing states across the entire submicron
diameter range. In this study, we developed a software which enables the DMA to scan at different
diameters for different time periods depending on their number concentrations. For example, DMA
scans at 100 nm last about 36 seconds, while scans at 700 nm last about 1.5 minutes, allowing for
enough particles to be sampled at larger particle sizes. The diameter set points (18 points) of DMA
scans are 100 nm, 120 nm, 160 nm, 200 nm, 235 nm, 270 nm, 300 nm, 335 nm, 370 nm, 400 nm, 435
nm, 470 nm, 500 nm, 535 nm, 570 nm, 600 nm, 635 nm, and 700 nm, with a full scan taking 20 minutes.

With this set-up, the particle number size distributions of BC-containing and BC-free aerosols

can be derived from the DMA-SP2 measurements using an inversion routine that mainly accounts for
the effects of the DMA transfer function and multiple charge. Assuming a BC density of 1.8 g/cm$^3$,



the BC volume equivalent diameter of each BC-containing particle (termed as BC core diameter, $D_c$)
can be calculated, assuming a core-shell structure. The particle number size distribution of BC-
containing aerosol, containing information of BC mixing states, can then be described using a two-
variable formulation $\frac{\partial N}{\partial \log (Dp) \partial \log (Dc)}$, and the multiple charge correction method proposed by Zhao et
al. (2021) was used here to account for the impacts of multiple charge on $\frac{\partial N}{\partial \log (Dp) \partial \log (Dc)}$ derivations.
Using the derived $\frac{\partial N}{\partial \log (Dp) \partial \log (Dc)}$, the BC mass size distribution with multiple charge corrections
accounted for can be derived through integrating rBC mass of each $D_c$. Details about the inversion
routines are introduced in Sect.  S1 of the supplement. Note that the effective density of bare BC or
BC contained in a particle can vary substantially due to BC morphology and existence of air voids
(Zhang et al., 2016;Zhao et al., 2020a). Therefore, a simple assumption of 1.8 g/cm$^3$ for BC density
could bring uncertainties to $D_c$ calculations. In addition, the optical equivalent diameter of BC-
containing aerosols cannot be retrieved in this study due to the failure of the SP2 split channel hardware
during the campaign, which rendered the leading-edge-only method unusable.
The average particle number size distribution (PNSD) derived from the DMA-SP2 system was
compared with the one obtained from independent SMPS measurements, as shown in Fig. 1a. The
detection limitations of the SP2 scattering channels (Raatikainen et al., 2017) caused the PNSD from
SP2 to be markedly lower than that from SMPS for diameters <200 nm. For diameters larger than 200
nm, the PNSD from SP2 was generally consistent with that from SMPS, with the average ratio of
SMPS to SP2 measurements being 0.89±0.05, which is similar to the phenomenon reported in
Raatikainen et al. (2017), with an average ratio of 0.82. Additionally, the observed rBC mass
concentrations correlated highly with the optically equivalent BC mass concentrations reported by the
AE33 aethalometer (R$^2$=0.96, and an average ratio of 0.96), as shown in Fig. S4. The consistency tests
between the DMA-SP2 system and SMPS measurements validated the number size distributions and
BC mass concentrations inverted from DMA-SP2 measurements.
The number fractions of BC-containing aerosols of various diameters can be calculated using the
DMA-SP2 measurements. Based on the time lag between the peak time of the scattering and the
incandescence signal (Schwarz et al., 2006;Moteki and Kondo, 2007;Sedlacek Iii et al., 2012), these
aerosols can be roughly divided into two categories: bare/thinly coated BC particles and thickly coated
BC particles. The time lag distribution of pure BC aerosols can be identified from the SP2 calibrations



using bare BC aerosols. Consequently, bare/thinly coated BC particles can be identified using the
calibrated critical lag-time.

**2.3 Q-ACSM measurements and positive matrix factorization (PMF) analysis**

The Q-ACSM measured non-refractory sub-micrometer (NR-PM$_1$) species, including organic
aerosol (OA), sulfate (SO$_4$), nitrate (NO$_3$), ammonium (NH$_4$), and chloride (Cl), at a time resolution
of 15 minutes. More detailed description can be referred to Liu et al. (2022) and Ng et al. (2011). The
mass spectra measured by the Q-ACSM were analyzed using ACSM standard data analysis software
(ACSM Local 1.5.10.0 Released July 6, 2015), written in Igor Pro (version 6.37). The composition-
dependent collection efficiency (CE) parameterization scheme proposed by Middlebrook et al. (2012)
was used to calculate the mass concentrations of OA and inorganic species. This was also detailed in
Liu et al. (2022). As calibration of the Q-ACSM was not available during this campaign, relative
ionization efficiencies (RIEs) of 5.15 and 0.7 for ammonium and sulfate from previous calibrations
were used, while the default RIEs of 1.4, 1.1, and 1.3 were used for organic aerosol, nitrate, and
chloride, respectively. The quality assurance of the Q-ACSM measurements was first performed
through comparing the mass concentrations of PM$_1$ (summation of measured NR-PM$_1$ concentrations
and rBC concentrations measured by the DMA-SP2 system) with PM$_1$ mass concentrations calculated
from the particle number size distribution measurements of SMPS, assuming an aerosol density of 1.6
g/cm$^3$. Good consistency (as shown in Fig. 1b) was achieved between SMPS and Q-ACSM
measurements (R=0.99), with NR-PM$_1$+rBC values slightly lower than PM$_1$ concentrations from
SMPS (an average ratio of 1.19). Two reasons explain the deviation: 1) the assumed average aerosol
density may be biased from the real variations; 2) some aerosol species are not measured by the Q-
ACSM, such as sub-micrometer dust.
Following the same procedure of the PMF analysis for the Q-ACSM measurements introduced in
Liu et al. (2022) and Zhai et al. (2022b), the PMF technique with the multilinear engine (ME-2)
(Canonaco et al., 2013;Canonaco et al., 2021) was applied to ACSM spectra for deconvolving OA into
different factors and detailed in the supplement. In total, four factors were identified, including two
primary OA (POA) factors: a hydrocarbon-like OA (HOA, O/C~0.16) and a cooking-like OA (COA,
O/C~0.14); and two oxygenated OA factors: a less oxidized oxygenated OA (LOOA, O/C~0.89) and
a more oxidized oxygenated OA (MOOA, O/C~0.94). SOA was represented by the summation of
LOOA and MOOA as done in previous studies (Kuang et al., 2020). The mass spectra of these factors,





the determination of the factor number, the selection of solutions and more details about the factor
analysis can be found in Sect. S2 of the supplement.

**3 Results and discussion**
**3.1 Overview of DMA-SP2 measurements and aerosol chemical composition**

During the observation period, the $PM_{2.5}$ mass concentration varied significantly (from 1 to 126

µg/m³) with an average of 20 µg/m³ and several pollution episodes were observed during relatively
stagnant conditions when wind speeds were near or below 1 m/s. The time series of meteorological
parameters as well as $PM_{2.5}$, ammonium sulfate (AS), ammonium nitrate (AN), SOA, HOA, COA and
rBC are shown in Fig. 2. The scheme proposed by Gysel et al. (2007) was used to identify AS and AN.
On average, secondary aerosols including nitrate, sulfate, ammonium and SOA together accounted for
about 80% of non-refractory $PM_1$ mass concentration and secondary aerosols increased substantially
during pollution episodes, demonstrating active secondary aerosol formations during the observations
which might significantly impact BC mass size distributions as well as BC mixing states. The average
air RH during the observations varied a lot from 42% to 98% with an average of 76%, suggesting that
the heterogeneous reactions that involve aerosol water were favored during this campaign, which is
consistent with the quick formation of ammonium nitrate in pollution episodes. On average, AN, AS
and SOA accounted for 33%, 25% and 42% of secondary aerosols respectively, which is consistent
with Zhai et al. (2022b) that nitrate concentrations are higher than sulfate during winter in Guangzhou
urban area, especially under pollution conditions. The time series of rBC mass concentrations were
shown in Fig. 2d, with rBC mass concentrations ranging from about 0.1 to 20 µg/m³ with an average
of 2.3 µg/m³. Resolved POA factors were HOA and COA, which is consistent with results of previous
studies in recent years that traffic emissions and cooking emissions are two main sources of primary
aerosols in Guangzhou urban area (Guo et al., 2020;Chen et al., 2021a;Chen et al., 2021b). The rBC




correlated highly with HOA ($R^2$=0.88), demonstrating that traffic emissions contributed dominantly to

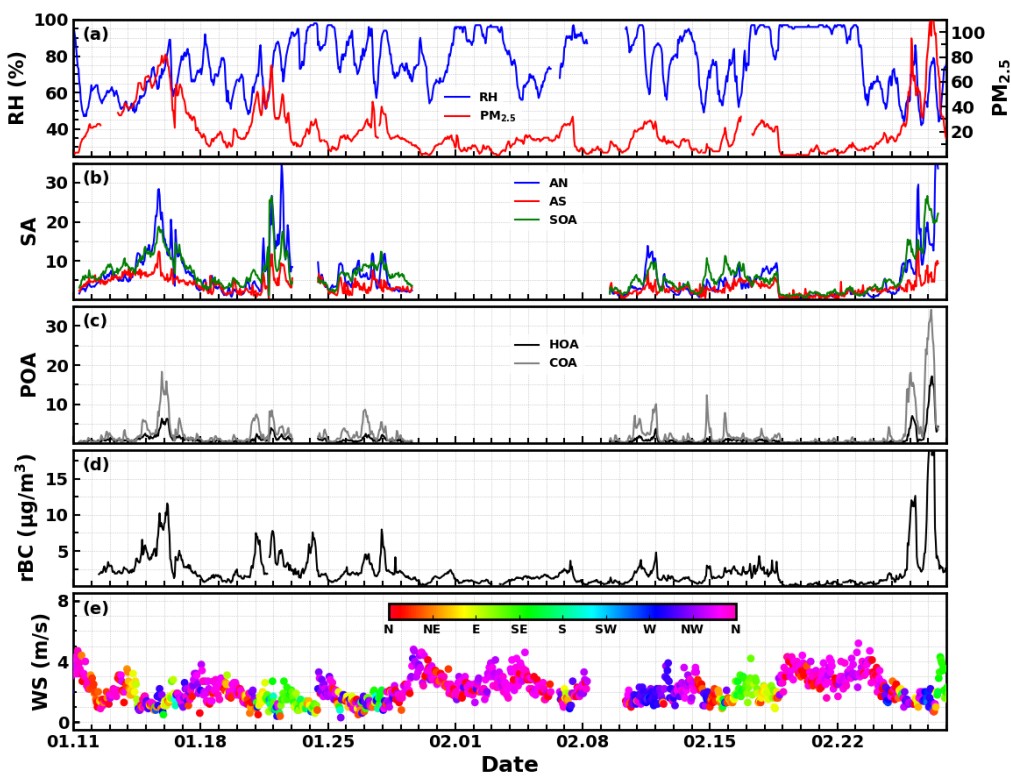

**Figure 2**. Timeseries of **(a)** RH and PM$_{2.5}$; **(b)** secondary aerosols including nitrate, sulfate and OOA; **(c)** HOA and COA; **(d)** rBC; **(e)** wind speed and directions.

atmospheric BC during the observations.

The observed average BC mass size distribution, as shown in Fig. 3a, exhibits a single lognormal

mode for diameters greater than 100 nm, with a fitted geometric mean ($D_{g,BC}$) of 258 nm ranging from
200 nm to 300 nm. The formula form of fitting is introduced in Eq.1 of the supplement, and the mean
of fitted geometric standard deviation ($\sigma_g$) is 1.69. A small mass mode might exist for diameters less
than 100 nm; however, it cannot be characterized due to the detection limitation of the SP2, which
measures BC-containing particles with a $D_c$ larger than 80 nm. Previous studies have reported BC mass
size distribution as a function of rBC core diameter (Kompalli et al., 2020;Liu et al., 2019). The
retrieved $D_{g,BC}$ is higher than most mass median diameters of rBC core measured in urban
environments, near 200 nm in urban Beijing (Liu et al. (2019) which is reasonable due to intrinsic
coatings (Adler et al., 2010). Some prior studies reported BC mass size distribution as a function of





mobility diameter $D_p$ measured by coupling the DMA with aethalometer (Stabile et al., 2012;Ning et
al., 2013;Zhao et al., 2019). A few studies also reported BC mass size distribution as a function of
aerodynamic diameter using the size-segregation filtering method (Hu et al., 2012). Zhao et al. (2019)
reported bimodal characteristics of BC mass size distribution in the North China Plain, with the second
mode accounting for most of the rBC mass and a $D_{g,BC}$ of the coarse mode ranging from 430 to 580
nm, which is much higher than the average one reported here. This difference may be attributed to the
markedly different sources of rBC (Zhang et al., 2020b) and the different roles of BC-containing

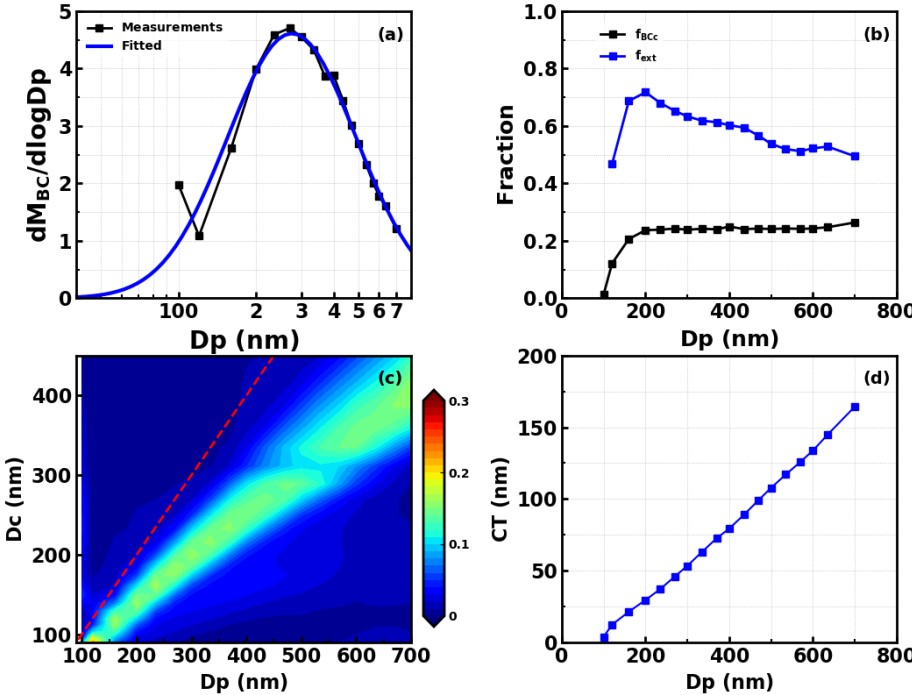

Figure 3. (a) Observed average BCMSD and the lognormal fitting curve; (b) Fractions of identified BC-containing aerosols in all aerosols at different diameters and fractions of externally mixed BC (bare BC) in BC-containing aerosols; (c) Average Dc distributions at different diameters, the red dashed line is the 1:1 line; (d) Average coating thickness (CT).

aerosols in the formation of secondary aerosols.

The BC mixing state is an essential factor for determining the BC's climate effects. Fig. 3b

displays the average size-dependent fractions of BC-containing aerosols ($f_{BCc}$) and the fraction of
identified externally mixed (bare/thinly coated) BC particles ($f_{ext}$) in all BC-containing aerosols. For





diameters below 200 nm, not all BC-free aerosols were detected by the SP2, so the number
concentration of all aerosols from SMPS measurements was used to calculate $f_{BCc}$. The results show a
decrease in $f_{BCc}$ from 200 nm to 100 nm, reaching nearly 0.01 at 100 nm. This may be due to the lower
fraction of BC-containing aerosols at smaller diameters, as well as the detection limit (~ 80 nm) of the
SP2, which may fail to detect many BC-containing aerosols below 80 nm. This can be further explained
based on the measurements from the volatility tandem differential mobility analyzer (V-TDMA) in
Guangzhou urban area in previous studies. Cheung et al. (2016) and Tan et al. (2016) used the number
fraction of remaining aerosols at 300 ℃ in V-TDMA measurements to represent $f_{BCc}$, assuming that
all BC-free aerosols had completely evaporated at this temperature. Their results showed an increasing
trend in $f_{BCc}$ from 0.62 to 0.86 for diameters ranging from 40 nm to 300 nm, which is much higher than
the values reported in this study (ranging from 200 nm to 700 nm, with an average of 0.24). Both
methods may be biased in $f_{BCc}$ measurements due to the detection limit of BC mass in the SP2, which
may underestimate $f_{BCc}$ (Zhao et al., 2020b), and the assumption that all BC-free aerosols have
evaporated at 300 ℃ in V-TDMA may overestimate $f_{BCc}$ by miscounting some aerosols with extremely
low volatility components (Tasoglou et al., 2020). Nevertheless, the low $f_{BCc}$ values obtained from
previous V-TDMA measurements confirm that $f_{BCc}$ is smaller for smaller diameters (Dp<200 nm). The
size-dependent $f_{BCc}$ is critical for simulating aerosol optical properties (Li et al., 2019) and CCN
predictions (Ren et al., 2018). The facts that most BC masses reside in particles larger than 100 nm
and most rBC masses would be detected by SP2 suggest that the $f_{BCc}$ reported from SP2 measurements
are more suitable for use in aerosol optical simulations. As for $f_{ext}$, the average $f_{ext}$ shows a decreasing
trend from 200 nm to 700 nm, with an average of 0.59, and the $f_{ext}$ at 100 nm being significantly
affected by the detection limit. This suggests that BC is generally externally mixed during the
observations, with a higher degree of aging for larger particles. The average distributions of rBC core
at different diameters are shown in Fig.3c, with a single mode at all diameters, deviating more from
the 1:1 line at larger diameters, again indicating a higher degree of aging for larger particles. The
estimated average coating thickness (CT) at different diameters is shown in Fig. 3d, with CT increasing
from 29 nm at 200 nm to 164 nm at 700 nm, and a relative coating thickness (RCT, Dp/Dc) ranging
from 1.27 at 200 nm to 1.88 at 700 nm. The RCT of 200 nm is similar to that of BC-containing aerosols
freshly emitted from diesel vehicles (Zhang et al., 2020b), consistent with diesel vehicle emissions
being the dominant source of BC traffic emissions (Bond et al., 2013).



The average diurnal variations of POA, rBC and secondary aerosol components including SOA,

AN and AS are depicted in Fig. 4a. In the morning, rBC and POA (HOA+COA) decrease due to

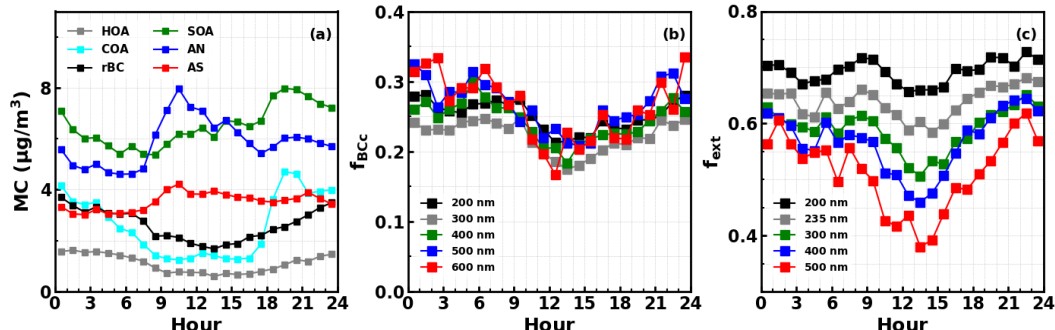

**Figure 4**. Average diurnal variations of **(a)** POA, SOA, ammonium nitrate (AN), ammonium sulfate (AS) and rBC ; **(b)** $f_{BCc}$ at different diameters; **(c)** $f_{ext}$ at different diameters.

dilution effects associated with boundary layer development, whereas SOA concentrations increase
since 08:00. However, POA and rBC begin to increase after 14:00, with the diurnal pattern of rBC
being generally consistent with that of HOA. The rapid increase in COA after 17:00 does not lead to a
significant rise in rBC, confirming that activities associated with cooking contribute negligibly to BC
emissions. AN and AS begin to decrease after noon, with SOA continuing to increase until 20:00,
which is in line with the findings reported in Zhai et al. (2022b) that the highest SOA mass
concentrations result from the coordination of daytime and nighttime SOA formation. The substantial
decrease in $f_{BCc}$ was observed in the morning when the prominent SOA formation occur, with the
decrease in $f_{BCc}$ being greater as the particle size increases, suggesting that secondary aerosols are
formed more efficiently on larger BC-free particles, which are then migrated to larger sizes. This is
further supported by the diurnal variations of $f_{ext}$, which revealed that secondary aerosol formation is
more efficient in larger BC-containing aerosols (Fig. 4c). A decrease in $f_{ext}$ from the morning to the
afternoon was most prominent for aerosols at 500 nm (0.17), while a small decrease in $f_{ext}$ (0.05) was
observed at 200 nm. This is consistent with the findings from the coating thickness results in Fig. 3d,
which showed that larger particles have higher coating thickness, and are therefore likely to contain
more aerosol water, thus favoring secondary aerosol formation via multiphase reactions.



### 3.2 Impacts of primary emissions and secondary aerosol formation on BC mass size distributions

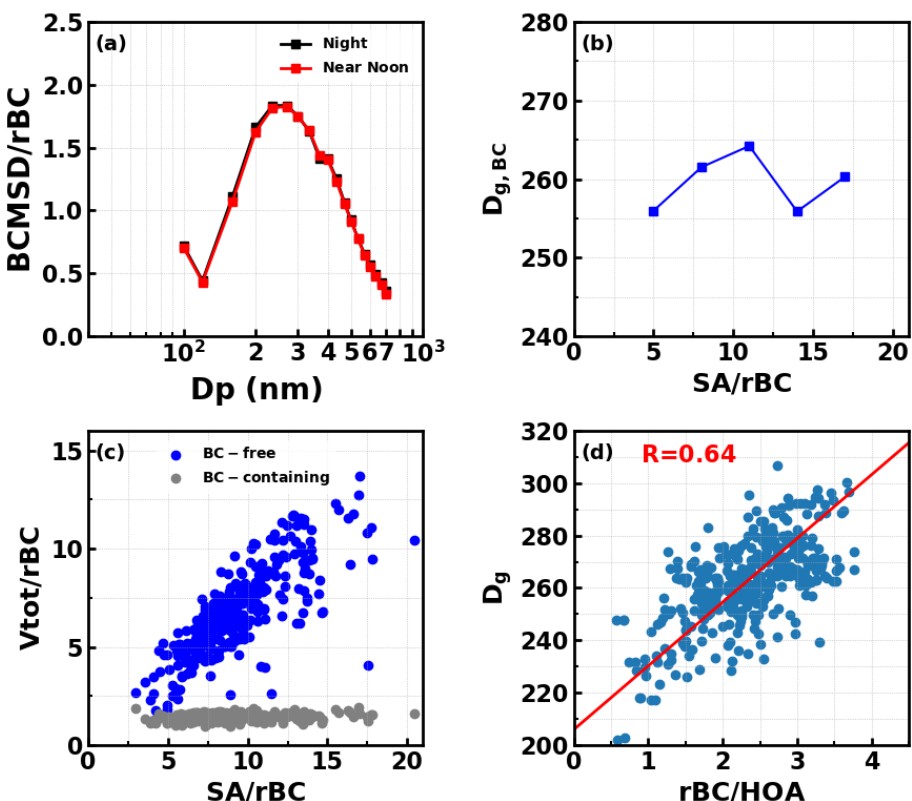

Figure 5. (a) Normalized average BCMSD during night and daytime; (b) Variations of Dg as a function of the ratio SA/rBC; (c) Scatter plots of SA/rBC and the ratio total volume (Vtot) to rBC of BC free and BC containing aerosols; (d) Correlations between Dg and SA/rBC variations during night.

The diurnal variations of $f_{ext}$ revealed that secondary aerosols are formed on BC-containing aerosols, thus impacting mixing states, and this might also result in changes in BC mass size distribution. The average BC mass size distributions are shown in Fig. 5a, which are normalized with rBC during the night (from local time 20:00 to 06:00 the next morning) and during afternoon when active secondary aerosol formation is at its final stage and the impacts of accumulations of primary emissions are relatively small (local time 12:00 to 17:00). Fig.5a indicates that daytime secondary aerosol formation does not modify the shape of BC mass size distributions, which is confirmed by the variations of $D_{g,BC}$ as a function of SA/rBC ratio shown in Fig. 5b (where SA includes mass concentrations of sulfate, nitrate, ammonium and SOA). The DMA-SP2 measurements distinguish BC-free and BC-containing aerosols, allowing for the volume variations of BC-free and BC-containing



aerosols with contributions of secondary aerosol formation to be differentiated. As seen in Fig. 5c, as
SA/rBC increases, the total volume of BC-containing aerosols increases very slightly, while secondary
aerosol formation mainly adds mass to BC-free aerosols, explaining the same BC mass size distribution
shape during both daytime and nighttime. Many previous studies have demonstrated that BC can serve
as sites for heterogenous reactions (Khalizov et al., 2010) and may even promote secondary aerosol
formation, thereby playing a significant role in haze formation (Zhang et al., 2020a). Results from
Zhang et al. (2021) pointed out that BC promotes sulfate formation in the urban area of Guangzhou
during summer. Our results indicate that BC plays a minor role in haze formation in Guangzhou during
winter, which might have significant implications for haze formation mechanisms in this region.

The BC primary emissions and their subsequent aging in the air determine the observed BC mass

size distributions. The aforementioned little influences of secondary aerosol formation on BC mass
size distribution evolution suggest that primary emissions played a significant role in the observed
variations in the BC mass size distributions. Traffic emissions dominate BC emissions during the
observations as discussed before, and diesel vehicles contribute dominantly to BC emissions from
traffic activities (Bond et al., 2013). The results from previous studies indicate that the ratio of
elemental carbon (EC) to organic carbon (OC) changed significantly depending on the external factors
(Adler et al., 2010;Lu et al., 2012), such as vehicle type, engine load, and driving conditions, etc. This
ratio represents the emission conditions of diesel vehicles, which also influences the size distributions
of diesel exhaust particles (Lähde et al., 2011;Han et al., 2019). Here, we use the ratio rBC/HOA to
represent different emission conditions related to traffic activities and investigate the potential effects
of rBC/HOA variations on BC mass size distributions. To avoid the potential effects of secondary
aerosol formation and daytime evaporation of HOA due to dilution, only data points from nighttime
(from 18:00 to 06:00 the next morning) are used, the results of which are displayed in Fig. 5d. Our
results showed for the first time that variations of $D_{g,BC}$ are strongly correlated with rBC/HOA, and a
linear relationship of $D_{g,BC}= 34\times rBC/HOA+177$ can be derived, indicating that a larger particle
diameter of BC-containing aerosols is associated with a higher BC content. Even though rBC and HOA
during nighttime in this study are accumulated from different vehicle sources, the relationship between
$D_{g,BC}$ and rBC/HOA still holds, suggesting that the black carbon content might be used for
parametrizing BC mass size distributions in traffic-related emissions.



### 3.3 Impacts of primary emissions and secondary aerosol formation on BC mixing states

As introduced in Sect. 3.1, both $f_{BCc}$ and $f_{ext}$ decreased during daytime due to secondary aerosol

formation. Here, the variations of daytime (from 08:00 to 18:00) $f_{BCc}$ and $f_{ext}$ under different SA/rBC

conditions were directly investigated and shown in Fig. 6. The $f_{BCc}$ values ranging from 200 nm to 600

nm decreased from about 0.3 to around 0.175 as SA/rBC increased from 5 to 15 (Fig. 6a), highlighting

significant impacts of secondary aerosol formation on $f_{BCc}$. As discussed in Sect. 3.1, the decrease of

$f_{BCc}$ should be associated with the fact that secondary aerosols are formed much more quickly on BC-

free aerosols than on BC-containing aerosols, which is consistent with the conclusion in Sect. 3.2 that

secondary aerosol formation mainly adds mass to BC-containing aerosols. New particle formation also

increases the number concentration of BC-free aerosols; however, its impact is limited for aerosol

particles beyond 200 nm (Zhang et al., 2012). The evolution of particle number size distribution shape

beyond 200 nm is mainly associated with vapor condensation, although coagulation  also play a role

(Seinfeld and Pandis, 2016).

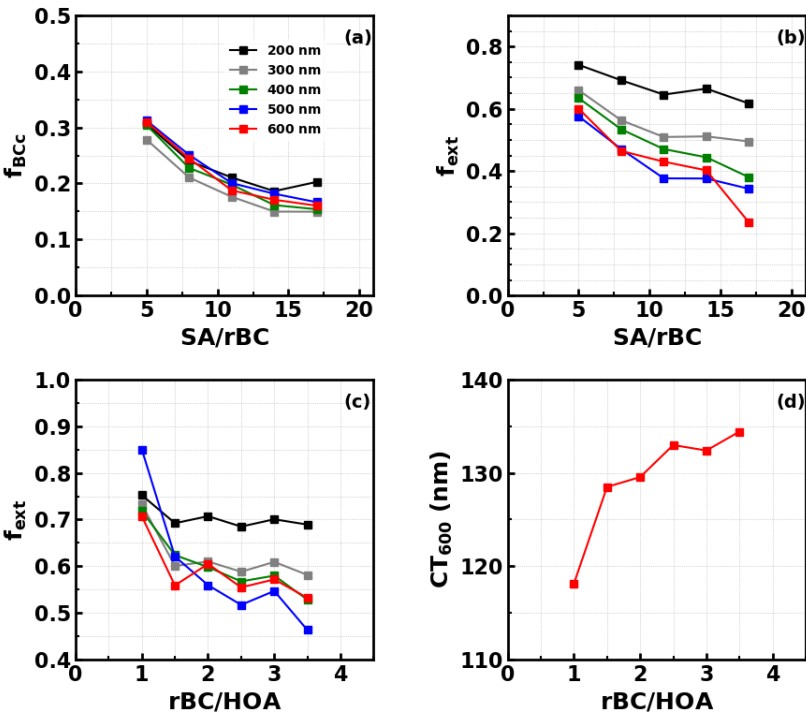

**Figure 6**. Variations of $f_{BCc}$ **(a)** and **(b)** $f_{ext}$ under different SA/rBC conditions; Variations of $f_{ext}$ **(c)** and coating thickness of aerosols with diameter of 600 nm **(d)** under different rBC/HOA conditions.



The variations in $f_{ext}$ under different SA/rBC conditions are presented in Fig. 6b. Larger particles
exhibited a more significant decrease in $f_{ext}$ with SA/rBC increasing from 5 to 17.5, for example, $f_{ext}$
of 200 nm particles only decreased from 0.75 to 0.6, while that of 600 nm particles decreased from 0.6
to 0.24. Aside from secondary aerosol formation, the emission conditions are also important factors
that influence BC mixing states, particularly due to the contribution of the co-emitted intrinsic organic
aerosols (Adler et al., 2010). At night, the shallow boundary layer facilitates the accumulation of
freshly emitted aerosols, leading to increases in the mass concentrations of rBC and POA (Fig. 4a) and
the mixing of freshly emitted and aged aerosols. Even so, the variations of $f_{ext}$ during night under
different rBC/HOA conditions might shed some lights on the impacts of primary emissions on BC
mixing states, which is shown in Fig.6c. As rBC/HOA increased from 1 to 3.5, $f_{ext}$ generally decreased,
especially for particles larger than 300 nm. The $f_{ext}$ of 600 nm decreased from 0.82 to 0.46, which is
higher than the degree of variations influenced by secondary aerosol formation as shown in Fig. 6b,
and most importantly SA/rBC decreased from 9.4 to 6.9 as rBC/HOA increased from 1 to 3.5,
demonstrating that the change of emission conditions has dominated the nighttime variations of BC
mixing states. The increased internal mixing degree of BC observed under higher rBC/HOA conditions
is also reflected in the variations of the average coating thickness of BC-containing aerosols. As seen
in Fig. 6d, the coating thickness of 600 nm BC-containing aerosols increased from 118 nm to 134 nm
as rBC/HOA increased from 1 to 3.5. Interestingly, as rBC/HOA increases, the relative amount of
coating to rBC becomes less, however, both fractions of internally mixed BC and coating thickness
increase, suggesting that a higher fraction of co-emitted intrinsic OA resides in BC-containing aerosols.
This section demonstrates the significant impacts of emission conditions of traffic sources on BC
mixing states.
**4. Atmospheric implications**
The evidence that the secondary aerosol formation mainly adds mass to BC-free particles suggests
that reactions occurring on or within BC-containing particles play a limited role in the formation of
haze in winter of Guangzhou. The urban area in Guangzhou is quite representative of most urban
regions where traffic emissions are the primary source of BC emissions. Hence, the above finding has
important implications for the haze formation mechanisms, particularly in Southern China where
primary aerosol emissions and meteorological conditions are similar to those in Guangzhou. The size
and mixing states of BC-containing particles determine their optical and hygroscopic properties, and



are therefore critical factors for evaluating the environmental, health, and climate effects of BC;
however, these factors are not adequately considered in both chemical transport and climate models
(Bond et al., 2013;Saleh, 2020). The finding that secondary aerosol formation has little effect on the
BC mass size distribution suggests that the study provides an excellent case scenario to investigate
how changes in traffic emissions affect the BC mass size distribution and mixing states. It further
shows that BC content in traffic emissions has a significant impact on both the BC mass size
distribution and mixing states, and the almost linear trends between $D_{g,BC}$ and rBC/HOA suggest that
BC content can be used as the key factor to parameterize both the BC mass size distribution and mixing
states from traffic emissions, which warrants future comprehensive investigation.
Traffic is a major global contributor to atmospheric BC concentrations, but other major sources,
such as biomass burning and coal combustion, as well as off-road diesel engines, also contribute
significantly to BC emissions and play an even more important role than traffic in some regions (Bond
et al., 2013). It is hence also important to investigate whether BC content of other major BC sources
than traffic are important in determining BC mass size distributions and mixing states. Saleh et al.
(2014) found that BC content has a major effect on the brownness of organic aerosols emitted from
biomass burning, and results from several later studies further confirmed this finding (Luo et al., 2022).
Recently, Saleh (2020a) discussed the potential for parameterizing the optical properties of brown
carbon using BC content in climate models. Moreover, the results of Luo et al. (2022) demonstrated
that BC content can also be used to parameterize the volume size distributions of aerosols emitted from
biomass burning, indicating that BC content plays an important role in both traffic emissions and
biomass burning emissions. These findings suggest that more comprehensive experiments should be
designed in the future to investigate the factors that control variations in BC mass size distributions
and mixing states, and to discuss how to use BC content from different major BC sources for
parameterizing BC mass size distributions and mixing states.








**Data availability**. All data needed are presented in time series of Figures and supplementary Figures,
raw datasets of this study are available from the corresponding author Ye Kuang (kuangye@jnu.edu.cn)
upon request.

**Competing interests**. The authors declare that they have no conflict of interest.

**Author Contributions**. YK and LL planned this campaign, with YK conceived and led this research.
FL performed the data analysis and wrote the manuscript together with YK. BL made the DMA-SP2,
SMPS and AE33 measurements together with FL and performed the post-processing of the SP2 data
with help of GZ, MMZ maintained the Q-ACSM during the observations and performed the PMF
analysis. JZ performed fund acquisition and supervision. TD, XJD and HBT helped the data acquisition
and revised the manuscript. All authors reviewed and edited the manuscript.


**Acknowledgments**
This work is supported by the Guangdong Major Project of Basic and Applied Basic Research (Grant
No. 2020B0301030004) ;Natural Science Foundation of Fujian Province (2021J01463); National
Natural Science Foundation of China (42175083 and 42105092); Guangdong Provincial Key Research
and Development Program (2020B1111360003); Guangdong Basic and Applied Basic Research
Foundation (2019A1515110791); The Special Fund Project for Science and Technology Innovation
Strategy of Guangdong Province (Grant No.2019B121205004).





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
