# Peer review of "Black carbon content of traffic emissions impacts significantly on black carbon"

_EGUsphere, 2023_

## Author Response (AR1)

**Reviewer#1**

**General Comment**

This manuscript presents results from a field campaign during which mas size distribution and composition of black carbon (BC) were measured using a coupling instrument between differential mobility analyzer and single-particle soot photometer (DMA-SP2). The impacts of BC content on the mass size distribution and mixing states were then investigated. Field measurements using DMA-SP2 for the BC mass size distribution and content are still very scarce and this study has made an important step towards understanding how BC content from traffic emissions can affect its mass size distribution and mixing states. The results show that although the BC content has important impacts on the BC mass size distribution and mixing states, secondary aerosol formation has little effect on the BC mass size distribution, hence exerting little forcing on the haze formation in Guangzhou. This provides important implications on the role of traffic emissions in haze formation and the control measures taken for air pollution prevention in megacities such as Guangzhou. The manuscript is well written and the results represent an important progress towards improving our understanding of the roles of BC in the radiative forcing and hence climate change. I would suggest a minor revision and a few issues need to be resolved.

**Response**: We appreciate for this comment which encouraged us to go further in the investigations of BC size distribution as well as mixing states of different sources and improve their representations in models.

**Specific comments**

**Comment**: A minor revision of the abstract is suggested to include the main implications of section 4.

**Response**: The following sentences are added:

"This study suggests that BC content can be used as the key factor to parameterize both the BC mass size distribution and mixing states from traffic emissions, which warrants future comprehensive investigation. In addition, other sources such as biomass burning and coal combustion also contribute substantially to BC emissions, it was important to investigate whether BC content of other major BC sources than traffic are also important in determining BC mass size distributions and mixing states. Overall, results of this study has significant implications for accurate representation of BC from different sources when modeling the impacts of BC. "

**Comment**: The overview seems to be too long. Splitting into two sections are suggested, for example, 3.1 Measurement overview, and 3.2 BC mass size distribution and mixing states. Then other sections need to be rearranged.

**Response**: Rearranged.

**Comment**: Caption of figure 5: (d) Correlations between Dg and rBC/HOA

during night.

**Response**: Corrected.

**Comment**: Line 107 on p4, SP2 (Raatikainen et al., 2017)

**Response**: Corrected.

**Comment**: What is the difference between Dg and Dg,BC? Please provide better clarification in the text.

**Response**: Dg is Dg,BC, all Dg is changed to Dg,BC in the revised manuscript.

**Comment**: Section 4 might include some conclusions of this study for better representation.

**Response**: The first paragraph of Section 4 is revised as:

"In this study, characterizations of BC mass size distributions and mixing state and their influencing factors were first investigated using measurements of DMA-SP2 system and aerosol mass spectrometer in Guangzhou urban area. Traffic emissions are the dominant source of atmospheric BC during the observations. The lognormal distribution represents well the BC mass size distribution, with the geometric mean varying between 200 and 300 nm with an average of 258±16 nm. On one hand, Tthe evidence that the secondary aerosol formation mainly adds mass to BC-free particles suggests that reactions occurring on or within BC-containing particles play a limited role in the formation of haze in winter of

Guangzhou. The urban area in Guangzhou is quite representative of most urban regions where traffic emissions are the primary source of BC emissions. Hence, the above finding has important implications for the haze formation mechanisms, particularly in Southern China where primary aerosol emissions and meteorological conditions are similar to those in Guangzhou. On the other hand, it was found that the daytime secondary aerosol formation reduced both fBCc and fext, with the decrease of fext being more pronounced for larger particles, suggesting that secondary aerosols actually formed on BC containing aerosols though their contribution to haze formation is small. The size and mixing states of BC-containing particles determine their optical and hygroscopic properties, and are therefore critical factors for evaluating the environmental, health, and climate effects of BC; however, these factors are not adequately considered in both chemical transport and climate models (Bond et al., 2013;Saleh, 2020). The finding that secondary aerosol formation has little effect on the BC mass size distribution suggests that the study provides an excellent case scenario to investigate how changes in traffic emissions affect the BC mass size distribution and mixing states. It further shows that BC content in traffic emissions has a significant impact on both the BC mass size distribution and mixing states, and the almost linear trends between $D_{g,BC}$ and rBC/HOA suggest that BC content can be used as the key factor to parameterize both the BC mass size distribution and mixing states from traffic emissions, which warrants future comprehensive investigation."

**General comments**

This manuscript investigated the mixing state of traffic emitted by BC in the Guangzhou urban area during winter. They used the DMA-SP2 setup to measure the size-resolved BC mixing state. The chemical composition of aerosols was measured with Q-ACSM. Their results suggested that traffic emissions can be a significant source of BC in the Guangzhou urban area. Their results suggested that the primary emission source significantly impacts BC mixing state and size distribution, and secondary aerosol might not play essential roles in BC deformation in the winter of Guangzhou. This manuscript is well-written, and the authors provide sufficient analysis to support their conclusions. This paper can improve our understanding of the secondary aerosol formation and BC mixing state and morphology evolution in the atmosphere. I have some comments which I hope can help improve this manuscript. Thus, I recommend a minor revision.

**Response**: We thank the reviewer for all valuable comments and suggestions which indeed improved this manuscript.

**Major comments**

**Comment**: I suggest adding error bars in all average plots (e.g., Figure 3-6) and adding a section of uncertainty calculation in SI.

**Response**: As suggested by the reviewer, error bars represented by standard deviations are added in some figures to deliver clearer the statistical results.

However, in some figures, if error bars were added, plots would be messed up which do not favor the clear depiction of the observed trend.

**Comment**: Have you accounted BC shape factor in your size distribution?

**Response**: The mobility diameter of DMA was used directly in this study, our measurements do not allow us to account for the BC shape factor due to the failure of the split channel.

**Comment**: For your AE33 measurements, I suggest adding details in the SI. You need to explain how did you derive eBC from AE33, how did you correct multi-scattering effects, etc.? Moreover, have you tried to estimate the fraction of BrC, which can also absorb NIR and IR, and thus, be classified as BC in AE33 and SP2?

**Response**: Details about the eBC derivations were added in the supplement. We did not estimate the fraction of BrC, because the BC concentrations are derived using the absorption measurements of 880 nm. Indeed, results of previous studies (Saleh, 2020;Yu et al., 2021) demonstrated that non-negligible BrC absorptions at near-infrared range, and results of Hoffer et al. (2017) demonstrated that absorption coefficient of tar balls at 880 nm is more than 10% of that at 470 nm. However, similar with those discussed in Luo et al. (2022), the BrC impacts on total aerosol absorptions at 880 nm would be less than 5% even on the basis of tar ball measurements which is actually not the case in this study, with traffic

emissions as well as secondary organic aerosol formations dominate BrC absorptions in Guangzhou urban area (discussions about this will be published in our another paper using year-long AE33 and aerosol chemical composition measurements). For SP2 measurements, the laser wavelength is 1064 nm which is longer than 880 nm, thus would be less affected.

**Comment**: Figure 2(a) shows the PM2.5 mass concentration. How did you measure that? I did not see you mention PM2.5 measurements.

**Response**: The following sentence is added in the manuscript:

"In addition, concentrations of PM2.5 and nitrogen dioxide (NO2) were obtained from China National Environmental Monitoring network which is publicly available (http://www.cnemc.cn/en/), there is a site located within 5 km distance to our observation site."

**Comment**: L233-235, "The average air … pollution episodes." Have you done any statistical analysis? This is unclear to me when I compare RH and secondary aerosol concentration by eyes.

**Response**: Yes, we reported these results on the basis of the statistical analysis. During pollution episodes, RH were not periods of near highest, however higher than 60% prevail.

**Comment**: L297-300, "In the morning, … of HOA." Do you have any boundary

layer measurements to support this? I would expect a lot of traffic emissions as the source of rBC and POC in the morning. Moreover, Why do POA and rBC increase after 14:00? What are their sources? That time is neither the traffic peak nor the cooking hour.

**Response**: We do not have boundary layer measurements to during the observations to support this directly, however, seasonal variations of boundary layer in this region have been discussed in previous studies (Yang et al., 2013). Note that diesel vehicle emissions contribute dominantly to rBC emissions, therefore, the emissions of rBC likely do not link tightly with traffic peak due to most of cars during traffic hours should be gasoline or electrical vehicles. Therefore, could expect that HOA variations also do not follow traffic peak. We thought the slight increase of rBC and HOA after 14:00 should be related with the weaker development of the boundary layer because of the weaker solar radiations especially during winter.

**Comment**: In the manuscript, you discussed SOA formed more efficiently on larger BC-free particles. SOA should be formed on particles with more surface area and activated pores in this case. Do you have any data to support this? Moreover, I think the primary source of SOA on BC is SOA condensation from the gas phase and increased particle size. Another question is the effect of new particle formation since it can also be dominated in smaller size regions (<100 nm) and decrease fBCc in that size region.

**Response**: Our results pointed out that secondary aerosols (SA) form much more efficiently on BC-free particles. As to SOA, its formation depends on the formation pathways, for formations associated with aerosol water (heterogeneous or aqueous), then surface area is not the important factor as discussed in (Kuang et al., 2020), our recent study about SOA formation in Guangzhou urban area demonstrate aerosol water might play significant roles in SOA formations (https://acp.copernicus.org/preprints/acp-2022-807/). We agree with the reviewer that new particle formations (if occurred) would play significant roles for particles in the nucleation and even Aitken mode (less than <100 nm). However, the focus of this study are BC mass size distributions and mixing states for diameters larger than 100 nm which is in the diameter range of condensation mode.

Reviewer#3

**General comments**

The manuscript by Li et al. performed a field campaign n Guangzhou urban area during winter to investigate the black carbon size distribution and mixing state from traffic emissions. Observations from DMA ad SP2 as well as other meteorological components are considered, and the results are well organized and discussed. It is found that the traffic emissions are the dominant source of atmospheric BC during the observations, and BC mass size distribution as well as its mixing properties are well studied. The paper is well organized and presented, and could be considered for publication after a revision, and I have only a few minor comments for the authors' consideration.

**Response**: Many thanks, your suggestions are very helpful.

**Major comments**

**Comment**: In the abstract session, the authors mostly list their conclusions directly from their observations and discussion. Would the authors also discuss the implementation of these observations? Such as, how would those observations benefit further BC modeling or their effects studies?

**Response**: The following sentences are added in the revised manuscript:

"This study suggests that BC content can be used as the key factor to parameterize both the BC mass size distribution and mixing states from traffic emissions, which warrants future comprehensive investigation. In addition, other sources

such as biomass burning and coal combustion also contribute substantially to BC emissions, it was important to investigate whether BC content of other major BC sources than traffic are also important in determining BC mass size distributions and mixing states"

**Comment**: Similar to the previous one, there are just a large number of conclusions from the observations, are they all newly noticed from this work, or some of them have been noticed before. The key points of this work are suggested to be better highlighted.

**Response**: The key findings of this work is that BC content impact significantly on BC mass size distributions and mixing states of traffic emissions which are both newly noticed.

**Comment**: In Figure 1b, although the correlation is high, but it seems that there is a systematic bias between the two observations, and would such systematic difference influence the conclusions?

**Response**:    The two observations should not be completely consistent, because SMPS measured size distributions of all aerosol components, however, some components such as dust were not detected by the ACSM which made the volume summations of NR-PM1 and rBC were slightly lower than those from SMPS measurements, which is consistent with previous observations.

**Comment**: Both $R^2$ and R are used in the manuscript to give the correlations

between different variables, and they are suggested to be unified.

**Response**: all R values are revised as $R^2$

**Comment**: Is it necessary to include a map to indicate the observational location of the campaign?

**Response**: This site locate in Guangzhou urban area, it was very easy to find where Guangzhou is, details of surroundings matter more which are given in Liu et al. (2022) and this sentence "Further details about this site can be found in Liu et al. (2022)" was already included in previous manuscript. The longitude and latitude of the observation site are given in the revised manuscript in case readers want to find precisely where the site is.

**Comment**: How significant are the uncertainties on the observations, and they should be briefly discussed or illustrated.

**Response**: The uncertainties of ACSM and SMPS was discussed in Sect 2.3.

Liu, L., Kuang, Y., Zhai, M., Xue, B., He, Y., Tao, J., Luo, B., Xu, W., Tao, J., Yin, C., Li, F., Xu, H., Deng, T., Deng, X., Tan, H., and Shao, M.: Strong light scattering of highly oxygenated organic aerosols impacts significantly on visibility degradation, Atmos. Chem. Phys., 22, 7713-7726, 10.5194/acp-22-7713-2022, 2022.

Hoffer, A., Tóth, Á., Pósfai, M., Chung, C. E., and Gelencsér, A.: Brown carbon absorption in the red and near-infrared spectral region, Atmos. Meas. Tech., 10, 2353-2359, 10.5194/amt-10-2353-2017, 2017.

Kuang, Y., He, Y., Xu, W., Yuan, B., Zhang, G., Ma, Z., Wu, C., Wang, C., Wang, S., Zhang, S., Tao, J., Ma, N., Su, H., Cheng, Y., Shao, M., and Sun, Y.: Photochemical Aqueous-Phase Reactions Induce Rapid Daytime Formation of Oxygenated Organic Aerosol on the North China Plain, Environmental science & technology, 54, 3849-3860, 10.1021/acs.est.9b06836, 2020.

Liu, L., Kuang, Y., Zhai, M., Xue, B., He, Y., Tao, J., Luo, B., Xu, W., Tao, J., Yin, C., Li, F., Xu, H., Deng, T., Deng, X., Tan, H., and Shao, M.: Strong light scattering of highly oxygenated organic aerosols impacts significantly on visibility degradation, Atmos. Chem. Phys., 22, 7713-7726, 10.5194/acp-22-7713-2022, 2022.

Luo, B., Kuang, Y., Huang, S., Song, Q., Hu, W., Li, W., Peng, Y., Chen, D., Yue, D., Yuan, B., and Shao, M.: Parameterizations of size distribution and refractive index of biomass burning organic aerosol with black carbon content, Atmos. Chem. Phys., 22, 12401-12415, 10.5194/acp-22-12401-2022, 2022.

Saleh, R.: From Measurements to Models: Toward Accurate Representation of Brown Carbon in Climate Calculations, Current Pollution Reports, 6, 90-104, 10.1007/s40726-020-00139-3, 2020.

Yang, D., Li, C., Lau, A. K. H., and Li, Y.: Long-term measurement of daytime atmospheric mixing layer height over Hong Kong, Journal of Geophysical Research, 118, 2422-2433, 2013.

Yu, Z., Cheng, Z., Magoon, G. R., Hajj, O. E., and Saleh, R.: Characterization of light-absorbing aerosols from a laboratory combustion source with two different photoacoustic techniques, Aerosol Science and Technology, 55, 387-397, 10.1080/02786826.2020.1849537, 2021.